:::: PLOS | ONE

# What drugs modify the risk of iatrogenic impulse-control disorders in Parkinson's disease? A preliminary pharmacoepidemiologic study

Nakyung Jeon[1]*, Marco Bortolato[2]*

**1** College of Pharmacy, Chonnam National University, Gwang-ju, Republic of Korea, **2** Department of Pharmacology and Toxicology, College of Pharmacy, University of Utah, Salt Lake City, Utah, United States of America

* nakyung.jeon@jnu.ac.kr (NJ); marco.bortolato@utah.edu (MB)

**Data Availability Statement:** Data were sourced from the Enterprise Data Warehouse (EDW), a database that houses electronic medical recordes of patients at the University of Utah Health System.

## Abstract

### Introduction

Parkinson's disease (PD) patients treated with pramipexole (PPX) and ropinirole (ROP) exhibit a higher risk of developing impulse control disorders (ICDs), including gambling disorder, compulsive shopping, and hypersexuality. The management of ICDs in PD is challenging, due to the limited availability of effective therapeutic alternatives or counteractive strategies. Here, we used a pharmacoepidemiological approach to verify whether the risk for PPX/ROP-associated ICDs in PD patients was reduced by drugs that have been posited to exert therapeutic effects on idiopathic ICDs–including atypical antipsychotics (AAs), selective serotonin reuptake inhibitors (SSRIs), and glutamatergic modulators (GMs).

### Methods

To quantify the strength of the associations between PPX/ROP and other medications with respect to ICD risk, odds ratios (ORs) were calculated by multivariable logistic regression, adjusting for age, gender, marital status race, psychiatric comorbidities, and use of cabergoline and levodopa.

### Results

A total of 935 patients were included in the analysis. Use of GMs, SSRIs, and AAs was not associated with a decreased ICD risk in PD patients treated with PPX/ROP; conversely, ICD risk was significantly increased in patients treated with either GMs (Adjusted Odds Ratio, ORa: 14.00 [3.58–54.44]) or SSRIs (ORa: 3.67 [1.07–12.59]). Results were inconclusive for AAs, as available data were insufficient to compute a reliable ORa.

### Conclusions

These results suggest that some of the key pharmacological strategies used to treat idiopathic ICD may not be effective for ICDs associated with PPX and ROP in PD patients.

To comply with University of Utah Health privacy and data security policies and regulatory constraints, only aggregated summary statistics and results of our analysis are permitted to share for publication. The authors have provided detailed results of the analyses in the paper. These restrictions are in place to maximize security of patient privacy and confidentiality. Access to these data can be granted to persons who go through EDW data access approval process, as the current authors did. To discuss the details of the approval process for the de-identified, raw dataset, please contact Nakyung Jeon, PhD (nakyung.jeon@jnu.ac.kr) or Vikrant Girish Deshmukk at Enterprise Datawarehouse (EDW) at the University of Utah (Vik.Deshmukh@utah.edu).

**Funding:** The author(s) received no specific funding for this work.

**Competing interests:** No authors have competing interests

Future studies with larger cohorts are needed to confirm, validate, and extend these findings.

## Introduction

A large body of evidence has documented that Parkinson's disease (PD) patients treated with dopamine replacement therapies (DRTs) exhibit a significantly greater risk of developing behavioral complications. [1,2] Pramipexole (PPX) and ropinirole (ROP), two of the best dopaminergic agonists for motor symptom management in early PD stages, [3,4] have been particularly associated with impulse control disorders (ICDs). [5–7] This umbrella term refers to a group of psychiatric conditions characterized by high risk-taking propensity, such as gambling disorder, compulsive shopping, compulsive sexual behavior, and binge eating. Although some of these entities have been moved to different diagnostic categories in the fifth edition of the Diagnostic and Statistical Manual for Mental Disorders (DSM-5) and the upcoming eleventh version of the International Classification of Diseases (ICD-11), this nomenclature is still largely used to refer to the impulsive/compulsive complications of DRTs in PD. [8]

ICDs often compound the severe social and financial burden experienced by PD patients, [1] and their management often poses serious clinical challenges. The best-validated therapeutic strategy for these problems is the dose reduction or discontinuation of PPX and ROP. [9] However, tapering these drugs is not always successful in alleviating ICDs, and can lead to the exacerbation of PD motor symptoms [10,11]. Furthermore, PPX/ROP discontinuation often leads to dopamine agonist withdrawal syndrome (DAWS), characterized by anxiety, panic attacks, diaphoresis, fatigue, dysphoria, and depression; these symptoms further impair functioning, cause significant distress, and are often refractory to other DRTs. [10] Unfortunately, no FDA-approved alternative treatments are currently available for these patients, underscoring the need for novel therapies to either prevent iatrogenic ICDs or mitigate their severity.

Over the past decade, several preliminary clinical trials have identified new putative therapeutic strategies for idiopathic ICDs and compulsive behaviors. Most of the research has pointed to five distinct categories of drugs that may have some efficacy as therapies for these disorders or related symptoms; selective serotonin reuptake inhibitors (SSRIs); [12–14] atypical antipsychotics (AAs); [15,16] mood stabilizers (MSs) such as lithium, carbamazepine, and valproic acid; [17,18] opioid receptor antagonists (ORAs); [19,20] and glutamatergic modulators (GMs), including amantadine, memantine, topiramate, and N-acetylcysteine have emerged as potential ICD treatments. [21,22] While some of these therapies have been anecdotally used in PPX/ROP-associated ICDs, no conclusive evidence is available on their efficacy in reducing these iatrogenic complications. [23,24]

Given the clinical and symptomatic similarities between ICDs secondary to PPX/ROP and their idiopathic counterparts, we hypothesized that the risk of drug-induced ICDs in PD patients may be mitigated by the same drugs that have been posited to exert therapeutic properties for idiopathic ICDs. To address this possibility, we performed a retrospective matched cohort study aimed at assessing whether each of the aforementioned drug classes may reduce the risk of ICD in PD patients treated with either PPX or ROP.

## Materials and methods

The present retrospective cohort study was conducted utilizing the institutional electronic medical records at the University of Utah (UU) Healthcare system. The UU Institutional Review Board approved this study by expedited review under a waiver of informed consent.

## Data source

Clinical and administrative data for 7,375 adult patients with at least one ICD 9/10-CM diagnostic code for PD (ICD-9-CM: 332, 332.0; ICD-10-CM: G20) between Oct 1994 and Feb 2019 were obtained from the University of Utah Enterprise Data Warehouse (UUEDW). The data contained diagnostic and prescription information from inpatient and outpatient records, along with demographic information.

## Inclusion/exclusion criteria and cohort entry

PD patients who received PPX or ROP at or after the first PD diagnosis were considered eligible. Cohort entry was defined as the date when a patient received the first PPX or ROP prescription. To ensure sufficient baseline and follow-up time to capture patient health information (including ICD development), we included only established patients who had one or more non-emergency hospital visits at least six months before and two years after the first PPX or ROP prescription. Patients with a diagnostic code for ICD (ICD-9-CM: 312.3X or ICD-10-CM: F63.X) at or before cohort entry were excluded.

## Identification of the initiation of GMs, SSRIs, AAs, MSs, and ORAs concomitant with PPX/ROP use

Prescription records of GMs, SSRIs, AAs, MSs and ORAs were identified to develop five individual cohorts and assess drug class effects on ICD risk individually. GMs included acamprosate, acetylcysteine, amantadine, memantine, and topiramate. SSRIs included citalopram, escitalopram, fluoxetine, fluvoxamine, paroxetine, and sertraline. AAs included asenapine, clozapine, quetiapine, olanzapine, paliperidone, iloperidone, lurasidone, risperidone, ziprasidone, and aripiprazole. MSs included lithium, carbamazepine, and valproic acid. Lastly, ORAs included naltrexone and nalmefene.

In each cohort, all patients were assigned to one of two groups defined by exposure to the five main drug classes. We excluded patients who had a prescription record of GMs, SSRIs, AAs, MSs, and ORAs prior to their cohort entry or did not continue to receive PPX/ROP at the time of the first exposure to the five drug classes.

## Index date, matching, and follow-up

For patients treated with GMs, SSRIs, AAs, MSs, and ORAs, the index date was defined as the first prescription date of these drugs. To establish a non-exposed group, for each exposed patient we selected up to 3 patients who never received the drug of interest, matching (with replacement) by age at cohort entry (± 1 year) and gender. Furthermore, we counted the number of days between the date of cohort entry (PPX/ROP initiation) and the index date (GM, SSRI, AA, MS and ORA initiation) for each exposed patient. Matching patients were followed up after the same number of days after the date of cohort entry to allow for the same duration of PPX/ROP treatment between groups at baseline. Baseline period was defined as the period between the cohort entry and the index date. Only non-exposed patients who were on PPX/ROP at the index datewere eligible for matching to an exposed patient.

## Analysis

We constructed three individual statistical models to assess associations between ICD incidence and GMs, SSRIs, and AAs. Of note, MSs and ORAs were only prescribed in less than 1% of the study population and the analyses of these two drug categories were not further pursued due to their limited statistical power. ICD incidence was estimated as the number of patients

with diagnostic code definitions for ICDs per 100 PD patients taking PPX/ROP who met inclusion/exclusion criteria.

To contrast the association of GM, SSRI, and AA use with ICD incidence, we estimated ICD incidences following exposure to each of the three drug classes among PPX/ROP receiving PD patients. The incidences were then compared with non-users of GMs, SSRIs, and AAs in separate models using conditional logistic regression models. Crude odds ratios (OR) along with 95% confidence intervals (CIs) were calculated to estimate the effect between the use of the psychiatric medications and the incidence of ICDs. We also adjusted the models by baseline characteristics if sample size allowed.[25] Patient characteristics considered for adjustment included: patient age at index date; gender; marital status; race; use of cabergoline or levodopa [26]; and psychiatric comorbidities (such as Delirium/Dementia/Amnestic/Other Cognitive Disorders, Depressive Disorders, Bipolar Disorders, Schizophrenia and Other Psychotic Disorders, and Alcohol or Substance-Related Disorders). [27–30] The comorbid conditions were identified by encounters with respective ICD–9/10–CM codes (available in S1 Table) up to 1 year prior to or at index date. The three main drug classes were not considered to adjust for each other. Variables were reported as percentages and means along with standard deviations (SD). Differences in baseline characteristics were assessed using the $X^2$-test, the exact calculation of the Fisher exact test, and 2-sample t-test as appropriate.

An alpha value of 0.05 was the significance level considered for all statistical tests. Analyses were generated using SAS software 9.4. Copyright, SAS Institute Inc. SAS and all other SAS Institute Inc. product or service names are registered trademarks or trademarks of SAS Institute Inc., Cary, NC, USA.

## Results

### Study population

Among 7,375 PD patients in the source study population, 1,791 patients (24.3%) had a PPX/ROP prescription at or after the first PD diagnosis. Of these, 940 (52.5%) patients had one or more non-emergency in- or outpatient visits at least six months before and two years after the cohort entry. Finally, 5 patients who had been diagnosed with ICDs at or before the first PPX/ROP prescription were excluded.

We assessed the utilization of the main five classes of drugs that have been associated with potential therapeutic effects in ICDs. Among the 935 patients included in our analysis, the most commonly prescribed drugs were GMs (505 patients, 54.0%) followed by SSRIs (473 patients, 50.5%) and AAs (237 patients, 25.3%). After excluding patients who were either prevalent users at cohort entry or no longer received PPX/ROP at index date, 264 (52.3%), 205 (43.3%), and 105 patients (44.3%) were remained in analyses who were newly prescribed with GMs, SSRIs, and AAs, respectively, at or after PPX/ROP initiation. Less than 1% of 935 patients initiated MSs or ORAs. As shown in Table 1, amantadine was predominant (227/264, 86.0%) among four types of the GMs considered in our analysis, followed by memantine (41/264, 15.5%). Among SSRI new users, es/citalopram (149/205, 72.7%), sertraline (61/205, 29.8%), and fluoxetine (29/205, 14.1%) were the most frequently prescribed. Lastly, 91.4% of AA new users received quetiapine (96/105). Olanzapine was the second most prevent prescriptions (19/105, 18.1%). Less than 5 patients received other AAs.

**GMs.** A total of 711 non-exposed patients were matched to the 264 exposed patients on age, gender, and duration of PPX/ROP. The ICD occurred at a significantly higher rate in the exposed group (20/264, 7.6%) compared to the non-exposed (10/711, 1.4%). The crude and adjusted OR were 4.99 (95% CI: 2.30–10.80) and 14.00 (95% CI: 3.58–54.4), respectively (Table 2).

**Table 1. Utilization of psychotropic drugs by class among 935 patients with Parkinson's disease initiating pramipexole/ropinirole therapy.** Please note that the total number of patients for each drug category is sometimes lower than the sum of all individuals listed for each treatment, since some patients were prescribed more than a single drug in each category.

| DRUG CATEGORIES | | Patients | % |
|---|---|---|---|
| **Glutamatergic modulators (GMs)** | Any GMs | 264 | 100.0% |
| | Amantadine | 227 | 86.0% |
| | Memantine | 41 | 15.5% |
| | Topiramate | 9 | 3.4% |
| | N-acetylcysteine | 9 | 3.4% |
| | Acamprosate | 0 | 0.0% |
| **Selective serotonin reuptake inhibitors (SSRIs)** | Any SSRIs | 205 | 100.0% |
| | Citalopram / Escitalopram | 149 | 72.7% |
| | Sertraline | 61 | 29.8% |
| | Fluoxetine | 29 | 14.1% |
| | Paroxetine | 10 | 4.9% |
| | Fluvoxamine | 2 | 1.0% |
| **Atypical antipsychotics (AAs)** | Any Aas | 105 | 100.0% |
| | Quetiapine | 96 | 91.4% |
| | Olanzapine | 19 | 18.1% |
| | Aripiprazole | 4 | 3.8% |
| | Risperidone | 3 | 2.9% |
| | Ziprasidone | 3 | 2.9% |
| | Clozapine | 1 | 1.0% |
| | Paliperidone | 1 | 1.0% |
| | Lurasidone | 0 | 0.0% |
| | Asenapine | 0 | 0.0% |
| | Iloperidone | 0 | 0.0% |
| **Mood stabilizers (MSs)** | Any MSs | 7 | 100.0% |
| | Lithium | 3 | 42.9% |
| | Carbamazepine | 3 | 42.9% |
| | Valproic acid/divalproex | 1 | 14.2% |
| **Opioid receptor antagonists (ORAs)** | Any ORAs | 0 | 0.0% |
| | Naltrexone | 0 | 0.0% |
| | Nalmefene | 0 | 0.0% |

**SSRIs.** After matching on age, gender, and duration of PPX/ROP at cohort entry, there were 554 patients in non-exposed group. Overall 23 patients (3.0%) developed ICD after initiation PPX/ROP. 13 cases (6.3%) were attributable to the concomitant use of SSRIs with PPX/ROP. Crude OR was 3.31 (95% CI: 1.44–7.65). The adjusted OR was 3.67 (95% CI: 1.07–12.60) after adjusting for comorbid mental disorder conditions, the use of levodopa/cabergoline, marital status, and race (Table 3).

**AAs.** After matching 105 exposed patients on age, gender, and duration of PPX/ROP at cohort entry, there were 295 patients in non-exposed group. In this AA model, ICD occurred at the lowest rate across all the three models, with 2.3% (9/400) of ICD incidence. The crude OR was 2.30 with 95% CI between 0.56 and 9.56. Adjusted OR could not be estimated due to problems with model convergence yielding unreliable OR estimate and confidence interval (Table 4).

**Table 2. Characteristics and impulse control disorder risk of glutamatergic modulators (GMs) users compared to non users.**

| | Overall (N = 975, 100%) | Non-users (N = 711, 72.9%) | Users (N = 264, 27.1%) | P-value[a] |
|---|---|---|---|---|
| **Characteristics** | | | | |
| Age, Mean (SD) | 66.8 (9.2) | 67.2 (8.9) | 65.8 (10.0) | 0.014 |
| Sex–Males (%) | 610 (62.6%) | 448 (63.0%) | 162 (61.4%) | 0.655 |
| Race—Caucasians (%) | 875 (89.7%) | 629 (88.5%) | 246 (93.2%) | 0.032 |
| Marital Status–Married (%) | 747 (76.6%) | 536 (75.4%) | 211(79.9%) | 0.148 |
| Levodopa / Cabergoline | 241 (24.7%) | 146 (20.5%) | 95 (36.0%) | <0.001 |
| **Comorbid Mental Illnesses** | | | | |
| Dementia / Cognitive Disorders | 149 (15.3%) | 122 (17.2%) | 27 (10.2%) | 0.007 |
| Depressive Disorders | 216 (22.2%) | 180 (25.3%) | 36 (13.6%) | <0.001 |
| Bipolar Disorders | 31 (3.2%) | 26 (3.7%) | 5 (1.9%) | 0.218 |
| Schizophrenia / Other Psychotic Disorders | 42 (4.3%) | 36 (5.1%) | 6 (2.3%) | 0.074 |
| Substance or Alcohol-Related Disorders | 24 (2.5%) | 21 (3.0%) | 3 (1.1%) | 0.160 |
| **Impulse Control Disorder Risk** | | | | |
| Number of events (%) | 30 (3.1%) | 10 (1.4%) | 20 (7.6%) | <0.001 |
| **Odds Ratios (95% Confidence Interval)** | | | | |
| Crude | | | 4.99 (2.30–10.80) | |
| Adjusted | | | 14.00 (3.58–54.44) | |

P-value[a]: $X^2$-test was performed to compare crude incidences between groups in terms of Impulse control disorder and baseline characteristics.

## Discussion

The main result of the present study was that GMs, SSRIs, and AAs were not found to statistically decrease the risk of ICDs associated with concomitant PPX/ROP use in PD patients. Conversely, we observed a positive association between GM use and ICD incidence in a cohort

**Table 3. Characteristics and impulse control disorder risk of selective serotonin reuptake inhibitors (SSRIs) users compared to non users.**

| | Overall (N = 759, 100%) | Non-users (N = 554, 73.0%) | Users (N = 205, 27.0%) | P-value[a] |
|---|---|---|---|---|
| **Characteristics** | | | | |
| Age, Mean (SD) | 23 (3.0%) | 10 (1.8%) | 13 (6.3%) | 0.014 |
| Sex–Males (%) | 67.6 (9.2) | 68.1 (8.7) | 66.3 (10.3) | 0.655 |
| Race—Caucasians (%) | 455 (60.0%) | 335 (60.5%) | 120 (58.5%) | 0.032 |
| Marital Status–Married (%) | 654 (86.2%) | 471 (85.0%) | 183 (89.3%) | 0.148 |
| Levodopa / Cabergoline | 591 (77.9%) | 434 (78.3%) | 157 (76.6%) | <0.001 |
| **Comorbid Mental Illnesses** | | | | |
| Dementia / Cognitive Disorders | 86 (11.3%) | 67 (12.1%) | 19 (9.3%) | 0.007 |
| Depressive Disorders | 161 (21.2%) | 99 (17.9%) | 62 (30.2%) | <0.001 |
| Bipolar Disorders | 26 (3.4%) | 24 (4.3%) | 2 (1.0%) | 0.218 |
| Schizophrenia / Other Psychotic Disorders | 47 (6.2%) | 42 (7.6%) | 5 (2.4%) | 0.074 |
| Substance or Alcohol-Related Disorders | 21 (2.8%) | 18 (3.3%) | 3 (1.5%) | 0.160 |
| **Impulse Control Disorder Risk** | | | | |
| Number of events (%) | 23 (3.0%) | 10 (1.8%) | 13 (6.3%) | 0.003 |
| **Odds Ratios (95% Confidence Interval)** | | | | |
| Crude | | | 3.31 (1.44–7.65) | |
| Adjusted | | | 3.67 (1.07–12.59) | |

P-value[a]: $X^2$-test was performed to compare crude incidences between groups in terms of Impulse control disorder and baseline characteristics.

**Table 4. Characteristics and impulse control disorder risk of atypical antipsychotics (AAs) users compared to non users.**

| | Overall (N = 400, 100%) | Non-users (N = 295, 73.8%) | Users (N = 105, 26.3%) | P-value[a] |
|---|---|---|---|---|
| **Characteristics** | | | | |
| Age, Mean (SD) | 9 (2.3%) | 5 (1.7%) | 4 (3.8%) | 0.250 |
| Sex–Males (%) | 70.2 (9.3) | 70.4 (8.9) | 69.6 (10.4) | 0.038 |
| Race—Caucasians (%) | 261 (65.3%) | 194 (65.8%) | 67 (63.8%) | 0.722 |
| Marital Status–Married (%) | 332 (83.0%) | 235 (79.7%) | 97 (92.4%) | 0.002 |
| Levodopa / Cabergoline | 302 (75.5%) | 225 (76.3%) | 77 (73.3%) | 0.597 |
| **Comorbid Mental Illnesses** | | | | |
| Dementia / Cognitive Disorders | 51 (12.8%) | 30 (10.2%) | 21 (10.0%) | 0.016 |
| Depressive Disorders | 49 (12.3%) | 28 (9.4%) | 21 (20.0%) | 0.049 |
| Bipolar Disorders | 8 (2.0%) | 2 (0.7%) | 6 (5.7%) | 0.005 |
| Schizophrenia / Other Psychotic Disorders | 16 (4.0%) | 5 (1.7%) | 11 (10.5%) | 0.0003 |
| Substance or Alcohol-Related Disorders | 7 (1.8%) | 4 (1.4%) | 3 (2.9%) | 0.385 |
| **Impulse Control Disorder Risk** | | | | |
| Number of events (%) | 23 (3.0%) | 10 (1.8%) | 13 (6.3%) | 0.003 |
| **Odds Ratios (95% Confidence Interval)** | | | | |
| Crude | | | 2.30 (0.56–9.56) | |
| Adjusted | | | Not performed | |

P-value[a]: $X^2$-test was performed to compare crude incidences between groups in terms of Impulse control disorder and baseline characteristics.

matched by gender, age at PPX/ROP initiation, and duration of PPX/ROP therapy (7.6% vs 1.4%, P<0.0001). Strikingly, this association remained after controlling for baseline characteristics, including levodopa/cabergoline use, psychiatric comorbidities, and patient demographics.

Given that amantadine accounted for 90% of GM prescriptions in this population, these findings appear to be in keeping with previous reports documenting that this drug may increase ICD risk in PD patients. [31,32] However, as the effects of amantadine are not exclusively underpinned by glutamate modulation but also reflect its dopamine-enhancing properties, [26] caution should be advocated against any generalized conclusion on this collective drug class.

Both crude and adjusted ORs suggested that SSRI users were at higher risk for ICD development. The lack of effectiveness of SSRIs in reducing the risk of iatrogenic ICDs in PD is surprising, in consideration of previous clinical reports documenting the efficacy of citalopram and escitalopram in improving impulsive behaviors in PD and psychiatric patients. [33–35] It is worth noting that the results of this analysis should not be considered conclusive, due to the observational nature of our study and the existence of residual confounding factors that could not be included in our analysis. However, considering the ICD incidence rate of 6.3% in SSRI user group was more than three times greater than non-SSRI users, the resulting increase in the absolute risk due to residual confounding factors would need to be substantial to reverse the association toward protective. Overall, the observed incidence rates provide some assurance regarding a limited potential for the role of SSRIs in reducing clinically significant ICD risk even if general concerns about bias are considered. If confirmed, the lack of efficacy of SSRIs in reducing PPX/ROP-associated ICDs may signify that the neurotransmitter changes induced by these antidepressants are unlikely to affect the activation of dopamine receptors induced by DRTs. Albeit preliminary, these findings collectively suggest that several key medications that have been postulated to exert therapeutic properties for idiopathic ICDs may not

have similar indications for their DRT-associated counterparts in PD. This discrepancy may reflect distinct pathophysiological underpinnings of these conditions. ICDs reflect the activation of $D_3$ receptors by drugs, rather than by dopamine, and as such may be less amenable to treatments that can modify dopaminergic release in the ventral striatum. In alignment with this idea, we recently showed that, in animal models, the effects of PPX on risky decision making were not based on modifications of dopamine release but were likely related to changes in downstream signaling of postsynaptic dopamine receptors. [36] It is also possible that ICDs in PD patients may feature specific characteristics secondary to the neurodegenerative condition, such as modifications of dopamine signaling in the ventral striatum secondary to hypofunction of the mesolimbic system in vulnerable PD patients. In support of this idea, recent studies have documented that dopaminergic deficits in the projections from the ventral tegmental area to the nucleus accumbens may be a critical predisposing factor to PPX/ROP-associated ICDs in PD. [37]

Several limitations of this study should be acknowledged. First, the prevalence of ICDs in our PD population was estimated at 3%, a rate markedly smaller than that reported in prior studies (6–60%); [2] this discrepancy may suggest that using diagnostic codes may have a low sensitivity in capturing patients with ICD. By the same token, however, this approach has high specificity, and therefore minimizes the likelihood of inclusion of patients without an accurate ICD diagnosis. Under these conditions, relative risk estimates are generally considered unbiased. [38] Nevertheless, given that we cannot estimate the rate of undiagnosed ICDs in this population, the extent of potential outcome misclassification remains unknown. An alternative explanation to account for this rate discrepancy may lie in the specific sociodemographic and cultural characteristics of Utah, which is the US state with the heaviest legal restrictions on gambling and betting. Thus, local PD patients may have fewer avenues to engage in these activities, ultimately resulting in lower ICD incidence.

Second, although our matching and adjustment analyses balanced patients on key risk factors -such as patient demographics, duration of PPX/ROP, and the use of levodopa/cabergoline -, the possibility of confounding bias cannot be entirely ruled out. For example, only 30% of patients treated with a SSRI had a comorbid diagnosis of depression at the time of SSRI initiation. Given that SSRIs are primarily prescribed for the treatment of depression and other psychiatric disorders, we anticipate a higher rate of undiagnosed depressive disorders in new SSRI users. An analogous argument could be used for amantadine, since this drug is used as a treatment for dyskinesias and motor fluctuations in PD, and these complications are robustly associated with ICDs. [39,40] However, our study controlled for PPX/ROP use in the study design and use of levodopa in regression model, which subsequently balanced patients with regard to dyskinesias.

Third, due to limitations in statistical power, we could not analyze other treatments for ICDs, such as MSs, ORAs, and voltage-dependent calcium channel inhibitors (e.g., pregabalin and gabapentin). [41] Furthermore, even the three main classes of drugs analyzed are rather heterogeneous in terms of mechanisms, leaving the possibility open that specific drugs in each category may have specific efficacy in reducing ICD severity. In the case of AAs, for example, our results were skewed by effects of quetiapine, which accounted for more than 90% of this group. While the use of this drug in PD is warranted by several reports showing its safety and antipsychotic efficacy in PD, [42–44] this drug is poorly effective in blocking $D_3$ dopamine receptors, [45,46] which are considered to play a key role in DRT-associated ICDs. [37] Conversely, AAs with relatively high affinity for $D_3$ receptors, such as risperidone [47], were under-represented in our sample, in alignment with warnings on the extrapyramidal adverse events associated with this drug. [48]

Finally, the small number of AA-treated patients developing ICDs in our population did not allow us to adjust for potential confounding factors. Future studies with adequate sample sizes will be needed to address these critical issues.

These caveats notwithstanding, the present findings suggest that GMs and SSRIs may not yield clinically meaningful therapeutic effects for DRT-associated ICDs in PD patients. This result awaits confirmation in causal analyses of observational data with adequate sample size, which will allow for controlling potential confounding factors while improving statistical power. For example, future analyses may be restricted to patients with indications for a specific drug and with comprehensive coverage of health care services, permitting a broader selection of variables for adjustment. It is also worth noting that the approach used in this study holds promise as a powerful data-mining methodology for drug discovery, which could be successfully coupled to screening in animal models of DRT-associated impulsivity. In this perspective, our group recently developed a rat model of probability discounting specifically optimized to capture the increase in probability discounting following PPX treatment. [36]

## Supporting information

**S1 Table. ICD 9/10 –CM codes by mental disorder type.**
(DOCX)

## Acknowledgments

The authors would like to thank Mike Newman at the University of Utah Enterprise Data Warehouse Team for providing de-identified electronic health data from UU Health.

## Author Contributions

**Formal analysis:** Nakyung Jeon.

**Methodology:** Nakyung Jeon.

**Supervision:** Nakyung Jeon, Marco Bortolato.

**Writing – original draft:** Nakyung Jeon, Marco Bortolato.

**Writing – review & editing:** Nakyung Jeon, Marco Bortolato.

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
