## [Decision Letter · Decision Letter 0]

27 Aug 2019

PONE-D-19-16461

What drugs modify the risk of iatrogenic impulse-control disorders in Parkinson’s disease? A preliminary pharmacoepidemiologic study

PLOS ONE

Dear Dr. Jeon,

Thank you for submitting your manuscript to PLOS ONE. After careful consideration, we feel that it has merit but does not fully meet PLOS ONE’s publication criteria as it currently stands. Therefore, we invite you to submit a revised version of the manuscript that addresses the points raised during the review process.

Please, be advised that submitting a revision does not guarantee acceptance.

We would appreciate receiving your revised manuscript by Oct 11 2019 11:59PM. To enhance the reproducibility of your results, we recommend that if applicable you deposit your laboratory protocols in protocols.io, where a protocol can be assigned its own identifier (DOI) such that it can be cited independently in the future. For instructions see: http://journals.plos.org/plosone/s/submission-guidelines#loc-laboratory-protocols

We look forward to receiving your revised manuscript.

Kind regards,

Vincenzo De Luca

Academic Editor

PLOS ONE

Journal Requirements:

2. Thank you for stating the following financial disclosure: No

a) Please provide an amended Funding Statement that declares *all* the funding or sources of support received during this specific study (whether external or internal to your organization) as detailed online in our guide for authors at http://journals.plos.org/plosone/s/submit-now.  

b) Please state what role the funders took in the study.  If any authors received a salary from any of your funders, please state which authors and which funder. If the funders had no role, please state: "The funders had no role in study design, data collection and analysis, decision to publish, or preparation of the manuscript."

Reviewers' comments:

Reviewer's Responses to Questions

**Comments to the Author**

1. Is the manuscript technically sound, and do the data support the conclusions?

Reviewer #1: Yes

Reviewer #2: Yes

2. Has the statistical analysis been performed appropriately and rigorously? 

Reviewer #1: Yes

Reviewer #2: Yes

3. Have the authors made all data underlying the findings in their manuscript fully available?

Reviewer #1: Yes

Reviewer #2: No

4. Is the manuscript presented in an intelligible fashion and written in standard English?

Reviewer #1: Yes

Reviewer #2: Yes

5. Review Comments to the Author

Reviewer #1: The present study investigates the potential of atypical antipsychotics (AAs), selective

35 serotonin reuptake inhibitors (SSRIs), and glutamatergic modulators in Parkinson’s disease (PD) patients treated with pramipexole (PPX) and ropinirole (ROP) to prevent impulse control disorders (ICDs). Results have the potential to enhance our undersanding of the respective side effects. However, two questions need to be considered before publication:

(i) drugs subsumized as antipsychotics are rather heterogeneous with respect to efficiency and potentail side effects. Quetiapine which was used in the majority of patients has profound anticholinergic side effects but a rather moderate antipsychotic efficiency. Other compounds such as clozapine were only received by some patients. These points should at least be mentioned when the efficacy of "antipsychotics" is discussed.

(ii) From a clinical standpoint cognitive impairment and dementia are common comorbidities in PD and may increase the vulnerability of patients towards side effects. Memantine is often prescribed in this situation.

Reviewer #2: 1) in the abstract specify the number of subjects and the results of the logistic regression

2) The figure with OR crude and adjusted can be eliminated and the info included in the text

3) Why did you include the table with ICD codes ? are those diagnosis from the medical records of the patients included in the study? if yes you should include also how many times that diagnosis has been present in your cohort

4) in tab 4 did you use chi-sq or logistic regression to test the effect of AP?

5) what covariate did you use to adjust for SSRI in tab 3 ; why did you not use the same covariates in the antipsychotic analysis?

6) how many glutamate modulators were administered in this cohort?

7) in tab 1 the number do not match if you consider monotherapy; were there patients taking 2 SSRI?

8) the alpha of 0.05 does not seem very conservative; comment on it.

9) were the patients in this cohort on anticolinergic meds?

10) Can you calculate specificity and sensitivity of the logistic regression model?

6. PLOS authors have the option to publish the peer review history of their article (what does this mean?). If published, this will include your full peer review and any attached files.

Reviewer #1: No

Reviewer #2: No

---

## [Author Response · Author response to Decision Letter 0]

30 Sep 2019

Response to Reviewer 1. 

Q1: The present study investigates the potential of atypical antipsychotics (AAs), selective serotonin reuptake inhibitors (SSRIs), and glutamatergic modulators in Parkinson’s disease (PD) patients treated with pramipexole (PPX) and ropinirole (ROP) to prevent impulse control disorders (ICDs). Results have the potential to enhance our understanding of the respective side effects. 

A1: We would like to thank the Reviewer for his/her kind comments. 

Q2:Drugs subsumed as antipsychotics are rather heterogeneous with respect to efficiency and potential side effects. Quetiapine, which was used in the majority of patients, has profound anticholinergic side effects but a rather moderate antipsychotic efficiency. Other compounds such as clozapine were only received by some patients. These points should at least be mentioned when the efficacy of "antipsychotics" is discussed.

A2:We agree with the Reviewer. In this revised version, this limitation has been clearly discussed (lines 326-341). The use of quetiapine in our population is likely in line with evidence showing that, unlike other antipsychotics, quetiapine has a good safety profile for PD patients. 

Q3:From a clinical standpoint, cognitive impairment and dementia are common comorbidities in PD and may increase the vulnerability of patients towards side effects. Memantine is often prescribed in this situation. 

A3:We agree with the Reviewer that PD patients with comorbid dementia or cognitive impairment are more likely to receive a memantine prescription and develop impulse control disorders. Thus, we included dementia/cognitive impairment as one of the covariates for the logistic regression models. 

Response to Reviewer 2. 

Q1:In the abstract, specify the number of subjects and the results of the logistic regression.

A1: As advised, we have added the number of patients included in our analysis (line 39), as well as the odds ratios computed from the logistic regression in the results section of the abstract (lines 41-44).

Q2:The figure with OR crude and adjusted can be eliminated and the info included in the text

A2: In agreement with this recommendation, we have removed the figure and corresponding text from the manuscript. 

Q3:Why did you include the table with ICD codes? Are those diagnoses from the medical records of the patients included in the study? If yes, you should include also how many times that diagnosis has been present in your cohort.

A3:The table includes the list of diagnosis codes that were considered to identify patients with a respective medical condition for covariate adjustment in our logistic regression. The use of the diagnoses listed in the table are explained on lines 166-170. 

Q4: In table 4, did you use chi-square or logistic regression to test the effect of AP?

A4: We used a logistic regression to test the effect of AP, as described on lines 159-161. 

Q5:What covariate did you use to adjust for SSRI in table 3? Why did you not use the same covariates in the antipsychotic analysis?

A5: As shown in table 3, age, sex, race, marital status, the use of levodopa/cabergoline, and several psychiatric conditions were included as covariates. We included the same covariates in the atypical antipsychotic (AA) analysis; however, there was an issue regarding model convergence yielding an unreliable estimate of the AA effect. For this reason, we did not report an adjusted odds ratio for the AA model. This was explained on lines 242-244. 

Q6: How many glutamate modulators were administered in this cohort?

A6: A total of 264 out of 935 patients received a glutamatergic medication such as amantadine, memantine, topiramate, N-acetylcysteine, and acamprosate. The break-down numbers of patients by type of glutamatergic medications are shown in Table 1. 

Q7: In Table 1, the numbers do not match if you consider monotherapy; were there patients taking 2 SSRI?

A7: Yes. There were patients who took two or more SSRIs or other drugs listed in the same category. This is now explained in detail in Table 1 legend. 

Q8: The alpha of 0.05 does not seem very conservative; comment on it.

A8: While we understand and appreciate this concern, our original decision to set alpha at P<0.05 was made in consideration of the preliminary, exploratory nature of these studies, which were specifically designed to identify potential negative associations between some drugs and PPX- and ROP-induced ICDs. At any rate, all p values were listed in the tables. Given that 1) use of corrections for multiple comparisons remains controversial with several authors, and 2) using a more conservative criterion for alpha would not affect our negative conclusions, we decided to maintain alpha at 0.05. 

Q9: Were the patients in this cohort on anticholinergic meds?

A9: Yes, but these treatments did not modify any of the associations reported in our study.

Q10: Can you calculate specificity and sensitivity of the logistic regression model?

A10: The use of logistic regression has a different implication for evaluating associations than they have in prediction modeling. Our study focuses on evaluation of associations between the use of a drug and the risk of impulse control disorders not developing a prediction model. Generally, in the evaluation of associations between an exposure and an outcome, specificity and sensitivity are not discussed. They are used to present model performance in prediction modeling.

---

## [Decision Letter · Decision Letter 1]

13 Dec 2019

What drugs modify the risk of iatrogenic impulse-control disorders in Parkinson’s disease? A preliminary pharmacoepidemiologic study

PONE-D-19-16461R1

Dear Dr. Jeon,

We are pleased to inform you that your manuscript has been judged scientifically suitable for publication and will be formally accepted for publication once it complies with all outstanding technical requirements.

With kind regards,

Vincenzo De Luca

Academic Editor

PLOS ONE

Additional Editor Comments (optional):

Reviewers' comments:

Reviewer's Responses to Questions

**Comments to the Author**

1. If the authors have adequately addressed your comments raised in a previous round of review and you feel that this manuscript is now acceptable for publication, you may indicate that here to bypass the “Comments to the Author” section, enter your conflict of interest statement in the “Confidential to Editor” section, and submit your "Accept" recommendation.

Reviewer #1: All comments have been addressed

2. Is the manuscript technically sound, and do the data support the conclusions?

Reviewer #1: Yes

3. Has the statistical analysis been performed appropriately and rigorously? 

Reviewer #1: I Don't Know

4. Have the authors made all data underlying the findings in their manuscript fully available?

Reviewer #1: Yes

5. Is the manuscript presented in an intelligible fashion and written in standard English?

Reviewer #1: Yes

6. Review Comments to the Author

Reviewer #1: All points raised by the reviewers have been adequately addressed by the authors. hence, I recommend the ms to be published in its present form.

7. PLOS authors have the option to publish the peer review history of their article (what does this mean?). If published, this will include your full peer review and any attached files.

Reviewer #1: No

---

## [Editor Report · Acceptance letter]

19 Dec 2019

PONE-D-19-16461R1 

What drugs modify the risk of iatrogenic impulse-control disorders in Parkinson’s disease? A preliminary pharmacoepidemiologic study 

Dear Dr. Jeon:

I am pleased to inform you that your manuscript has been deemed suitable for publication in PLOS ONE. Congratulations! Your manuscript is now with our production department. 

With kind regards,

on behalf of

Dr. Vincenzo De Luca 

Academic Editor

PLOS ONE